# An asymmetric nautilus-like HflK/C assembly controls FtsH proteolysis of membrane proteins

Alireza Ghanbarpour [1,2✉], Bertina Telusma [1], Barrett M Powell[1], Jia Jia Zhang [1], Isabella Bolstad[1], Carolyn Vargas[3,4,5], Sandro Keller [3,4,5], Tania A Baker[1], Robert T Sauer [1✉] & Joseph H Davis [1,6✉]

## Abstract

The AAA protease FtsH associates with HflK/C subunits to form a megadalton-size complex that spans the inner membrane and extends into the periplasm of *E. coli*. How this bacterial complex and homologous assemblies in eukaryotic organelles recruit, extract, and degrade membrane-embedded substrates is unclear. Following the overproduction of protein components, recent cryo-EM structures showed symmetric HflK/C cages surrounding FtsH in a manner proposed to inhibit the degradation of membrane-embedded substrates. Here, we present structures of native protein complexes, in which HflK/C instead forms an asymmetric nautilus-shaped assembly with an entryway for membrane-embedded substrates to reach and be engaged by FtsH. Consistent with this nautilus-like structure, proteomic assays suggest that HflK/C enhances FtsH degradation of certain membrane-embedded substrates. Membrane curvature in our FtsH•HflK/C complexes is opposite that of surrounding membrane regions, a property that correlates with lipid scramblase activity and possibly with FtsH's function in the degradation of membrane-embedded proteins.

**Keywords** Cryo-EM; Proteostasis; AAA Protease; Macromolecular Complexes
**Subject Categories** Microbiology, Virology & Host Pathogen Interaction; Post-translational Modifications & Proteolysis; Structural Biology

## Introduction

AAA proteases sculpt the proteome to specific biological needs by using energy derived from ATP hydrolysis to unfold and degrade damaged or unneeded proteins (Baker and Sauer, 2012). Among AAA proteases, FtsH is unusual in being membrane-anchored and thus positioned to degrade membrane proteins (Ito and Akiyama, 2005). FtsH homologs are widespread in eubacteria (Langklotz et al, 2012)

and eukaryotic organelles of bacterial origin (Yi et al, 2022), and have been suggested as targets for anti-microbial (Hinz et al, 2011) and antimalarial (Amberg-Johnson et al, 2017) therapeutics.

FtsH assembles into a homohexamer that is active in protein unfolding and degradation (Kihara et al, 1998; Tomoyasu et al, 1995). In *Escherichia coli*, each FtsH subunit consists of a transmembrane helix, a periplasmic domain, a second transmembrane helix, and a cytoplasmic AAA module and zinc-peptidase domain (Bittner et al, 2017; Ito and Akiyama, 2005) (Appendix Fig. S1). ATP hydrolysis within the AAA modules of FtsH powers substrate unfolding, extraction of membrane-embedded substrates from the lipid bilayer, and translocation of denatured polypeptides through an axial channel and into the peptidase chamber. FtsH degrades both membrane-embedded and cytoplasmic proteins (Bittner et al, 2017), and is one of several AAA proteases in *E. coli* that degrade ssrA-tagged proteins resulting from tmRNA-mediated ribosome rescue (Herman et al, 2003). Disordered peptide segments in these substrates are engaged by the axial channel of the FtsH hexamer, allowing for specific recognition and subsequent mechanical unfolding and degradation (Bittner et al, 2017; Ito and Akiyama, 2005).

In *E. coli*, FtsH partners with two additional proteins—HflK and HflC, which contain SPFH domains, forming a large holoenzyme that extends across the inner membrane and into the periplasm (Saikawa et al, 2004) (Appendix Fig. S1). Similar super-complexes bearing SPFH domains also exist in eukaryotic organelles (Fu and MacKinnon 2024). A recent cryo-EM structure of an FtsH super-complex from *E. coli* determined following overproduction of the protein components and application of fourfold symmetry (C4) during 3D reconstruction, revealed a cage consisting of 24 alternating subunits of HflK and HflC tightly enclosing four FtsH hexamers (Ma et al, 2022; Qiao et al, 2022). Paradoxically, however, this structure suggests that HflK/C prevents the degradation of membrane-embedded substrates by insulating them from the FtsH proteolytic machinery.

Here, we present markedly different FtsH•HflK/C super-complex structures, which also contain 24 HflK/C subunits but only one or two FtsH hexamers. Notably, the HflK/C subunits in these structures form an asymmetric assembly, similar to a nautilus shell, with a passageway to the interior of the shell. These nautilus-like complexes were purified

[1]Department of Biology, Massachusetts Institute of Technology, Cambridge, MA 02129, USA. [2]Department of Biochemistry and Molecular Biophysics, Washington University in St. Louis, St. Louis, MO 63110, USA. [3]Biophysics, Institute of Molecular Biosciences (IMB), NAWI Graz, University of Graz, Humboldtstr. 50/III, Graz 8010, Austria. [4]Field of Excellence BioHealth, University of Graz, Graz, Austria. [5]BioTechMed-Graz, Graz, Austria. [6]Program in Computational and Systems Biology, Massachusetts Institute of Technology, Cambridge, MA 02129, USA. ✉E-mail: alirezag@wustl.edu; bobsauer@mit.edu; jhdavis@mit.edu

without protein overproduction using an affinity tag added to chromosomally encoded FtsH. Structures with similar topology were obtained after detergent solubilization or after detergent-free extraction using a nanodisc-forming polymer. The lipid domains in these structures display unexpected curvature, which correlates with enhanced rates of lipid scrambling. As such scrambling has been linked to a thinned membrane, this activity could aid FtsH in extracting and degrading membrane-embedded substrates. In addition, we performed cell-wide proteomic experiments that revealed that several previously reported membrane-embedded protein substrates of FtsH are present at higher levels in the absence of HflK/C, supporting a model in which a nautilus-like FtsH•HflK/C holoenzyme degrades these substrates.

# Results

## FtsH•HflK/C purification without protein overexpression

We engineered an *E. coli* strain in which the chromosomal *fts*H gene was modified to encode a C-terminal FLAG epitope and then used anti-FLAG antibody resin to purify tagged FtsH and associated HflK and HflC (Appendix Fig. S2A; see "Methods"). Addition of the FLAG tag did not slow *E. coli* growth at 37 °C (Appendix Fig. S2B), despite FtsH protease activity being essential for cell viability (Herman et al,

1993). Thus, we inferred that the tag did not markedly affect activity in cells. In preparation for single-particle cryo-EM, we used *n*-dodecyl-β-D-maltoside (DDM) or glyco-diosgenin (GDN) detergents to solubilize the FtsH•HflK/C complex prior to purification. Following DDM solubilization, the purified FtsH•HflK/C complex was active in ATP hydrolysis and ATP-dependent protein degradation (Appendix Fig. S2C,D).

## Structural features of an asymmetric FtsH•HflK/C complex

We refined a ~4.4 Å resolution cryo-EM structure of the DDM-solubilized complex without imposed symmetry, revealing an asymmetric HflK/C complex resembling a nautilus shell with two internal FtsH hexamers (Fig. 1A–C; Appendix Figs. S3–4). A similar cryo-EM structure, albeit at lower resolution, was obtained when the complex was solubilized with the detergent GDN (Appendix Fig. S5), indicating that the nautilus-like structure was not unique to a specific detergent. Moreover, this nautilus-like structure was not the result of preferential orientations or preferential positioning at the air-water interface, as the 3D-FSC (Tan et al, 2017) measure of sphericity was 0.89 with diverse projection angles observed (Appendix Fig. S4C,D). Furthermore, tomographic reconstructions revealed random particle orientations on the EM grid (Appendix Fig. S6).

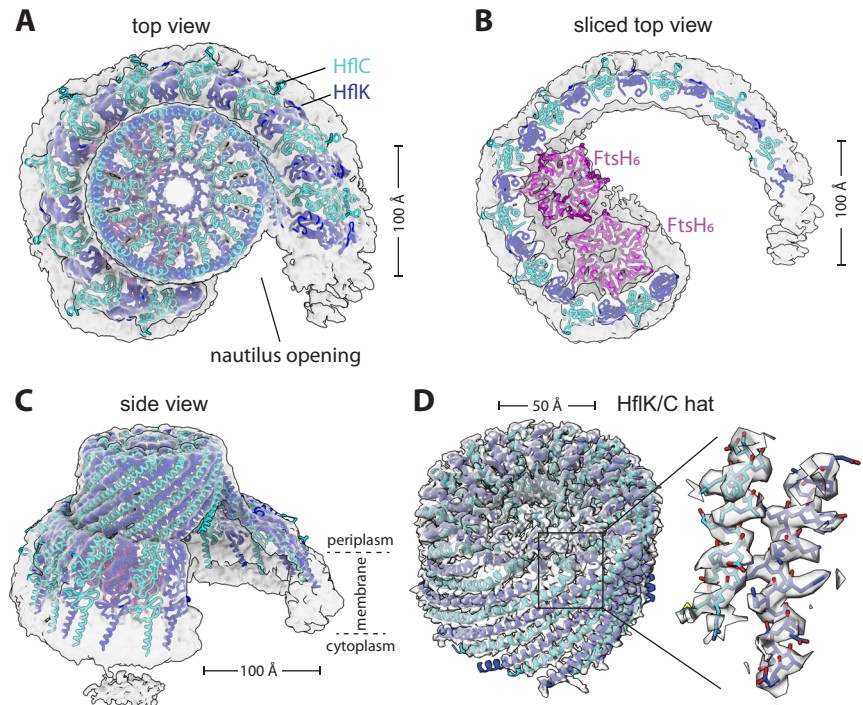

**Figure 1. Nautilus-like structure of a FtsH•HflK/C super-complex.**

(A) Density map (unsharpened) and cartoon models of HflK (blue) and HflC (cyan) viewed from top. Nautilus opening into the HflK/C chamber noted. Map was resolved to a GS-FSC resolution of ~4.5 Å. (B) A sliced top view of the complex highlighting two FtsH hexamers (pink) interacting with SPFH domains of HflK within the nautilus chamber. (C) A side view of the nautilus super-complex, following representation from (A). Approximate location of the membrane depicted with periplasmic and cytoplasmic faces of the inner membrane noted. (D) A hat-like portion of the structure locally refined to ~3.5 Å GS-FSC resolution, color schemes as in (A). This map allowed for clear identification of HflK and HflC. The inset displays is a sharpened map and the model built using this map, highlighting bulky residues in helical regions of HflK and HflC.

HflK and HflC share sequence and structural homology and were difficult to distinguish in the 4.4-Å density map of the DDM-solubilized complex. However, after signal subtraction and local refinement without symmetry (see "Methods"), we refined the periplasmic portion of the nautilus structure farthest from the membrane, which resembles a hat, to 3.5-Å resolution (Appendix Figs. S7 and 8). This structure allowed visualization of side chains and unambiguous assignment of ~110 C-terminal residues of 24 alternating HflK and HflC subunits (Fig. 1D). The corresponding hat region of the C4-symmetric HflK/C structure (pdb 7WI3) (Qiao et al, 2022) aligned to the asymmetric nautilus hat with an RMSD of 2.3 Å for the 2545 common Cα positions. SPFH domains, present in each HflK and HflC subunit, formed the membrane-proximal sides of the nautilus chamber, but only 21 of these 24 domains could be modeled (Figs. 1 and 2). The entryway into the nautilus chamber was located near the "missing" SPFH domains, which presumably adopted multiple conformations, or were proteolytically truncated. The final model, which had good geometry (Table 1), included residues 79–355 for most HflK subunits, residues 1–160 and 191–329 for most HflC subunits, and residues 31–97 of the periplasmic domains of two FtsH hexamers.

The nautilus chamber in our structure contained two hexamers composed of FtsH periplasmic domains, with one hexamer better resolved than the other. In their closed C4-symmetric structure, Qiao et al identified contacts between Arg[191] of HflK and residues 62-64 of the periplasmic domain of FtsH. Although the resolution of our asymmetric nautilus structure was not sufficient to confidently assign these specific contacts, the backbones in both structures were positioned similarly (Figs. 1B and 2). Because the tightly closed HflK/C cage in the C4-symmetric structure (Qiao et al, 2022) contained four FtsH hexamers, whereas our open nautilus chamber contained fewer FtsH hexamers, we hypothesized that chamber curvature might be proportional to the number of internal FtsH hexamers. To test this hypothesis, we performed focused classification, using a mask encompassing the periplasmic hexamer of FtsH with lower occupancy (Appendix Fig. S3). After 3D classification and refinement of the classified particles, assemblies with either one or two FtsH hexamers were evident

(Appendix Figs. S9 and S10). A comparison of these maps (Fig. 2A,B) revealed no significant structural rearrangements in the nautilus shell, indicating that chamber curvature does not simply increase with the number of enclosed FtsH hexamers. Moreover, when we overproduced HflK and HflC prior to affinity purification via a FLAG-tag on HflC, the resulting structure (Fig. 2C; Appendix Figs. S11 and S12) had a nautilus-like HflK/C chamber lacking FtsH density, but with curvature similar to our structures with one or two FtsH hexamers. In each instance, our maps were clearly distinct from the C4-symmetrized structure from Qiao et al (Fig. 2D).

We next searched for structural heterogeneity in our DDM-solubilized FtsH•HflK/C dataset using cryoDRGN (Kinman et al, 2023; Sun et al, 2023; Zhong et al, 2021). This analysis revealed the expected nautilus-like structures with either one or two FtsH hexamers and additional differences in the width of the entryway into the nautilus chamber (Movie EV1), but closed HflK/C structures or complexes with four FtsH hexamers were not observed.

## Unusual curvature of the membrane microdomain of FtsH•HflK/C

The density map of DDM-solubilized FtsH•HflK/C showed the approximate positions of lipid head groups in the bilayer (Appendix Fig. S4), but individual lipid molecules could not be resolved or modeled, and the density may represent a mixture of detergent and lipids. Notably, however, the bilayer curved inward within the complex, whereas the bacterial inner membrane as whole curves outward (Khanna and Villa, 2022). This unusual membrane geometry was not a function of detergent solubilization of the complex, as similar curvature was observed in a lower-resolution (~12 Å) 3D reconstruction of the affinity-purified FtsH•HflK/C complex following extraction with the nanodisc-forming polymer Carboxy-DIBMA, and native lipids (Fig. 3A; Appendix Figs. S13 and S14). Indeed, a 2D-class average side view of these particles showed pronounced inward membrane curvature within the complex and reversal of this curvature as the membrane extended beyond the embedded FtsH•HflK/C complex (Fig. 3B).

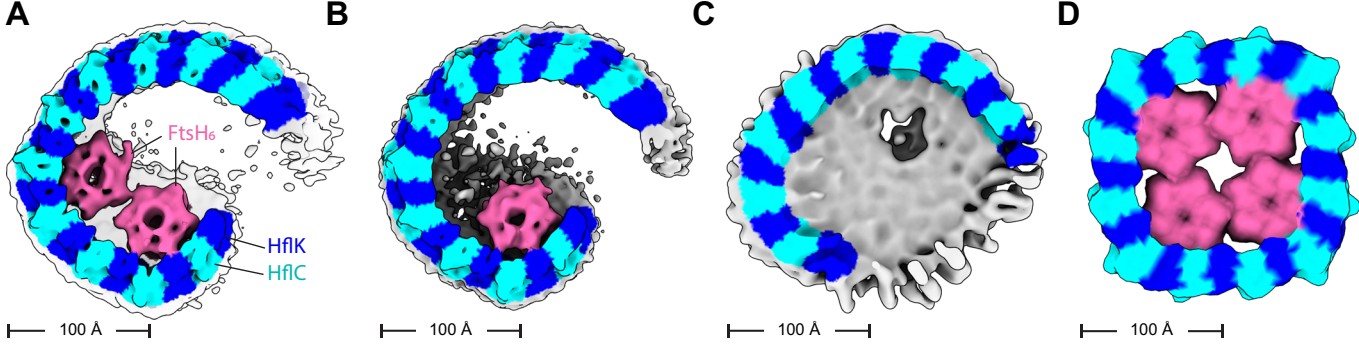

**Figure 2. HflK/C structures with different numbers of FtsH hexamers.**

Density maps were low-pass filtered to 10 Å and colored using the underlying atomic model (HflK—blue; HflC—cyan, FtsH—pink; unassigned membrane or detergent—gray). (A) Two FtsH hexamers per super-complex as refined in Appendix Fig. S10. (B) One FtsH hexamer per super-complex as refined in Appendix Fig. S9. (C) FtsH-free super-complex affinity-purified from cells overexpressing HflC-FLAG and HflK, and refined as depicted in Appendix Figs. S11 and S12. (D) C4-symmetric structure with four FtsH hexamers (Qiao et al, 2022).

**Table 1. Cryo-EM data collection, processing, model building, and validation statistics.**

| Sample name | FtsH•HflK/C map a (DDM extracted) | FtsH•HflK/C hat-like map (locally refined) | FtsH•HflK/C map I | FtsH•HflK/C map II | HflK/C | FtsH•HflK/C (Carboxy-DIBMA) |
|---|---|---|---|---|---|---|
| Description | Primary map analyzed | Locally refined map focused on HflK/C 'hat' | Classified map a with single FtsH$_6$ | Classified map a with two FtsH$_6$ | HflK/C overexpressed | Detergent-free solubilized sample |
| PDB code | 9CZ2 | 9CZ1 | n/a | n/a | n/a | n/a |
| EMDB ID | 46057 | 46056 | 46061 | 46059 | 46062 | 46058 |
| EMPIAR code | 12472 | | | | 12471 | — |
| Defocus range (μm) | −0.05 to −1.75 | | | | −0.5 to −2 | −0.30 to 1.75 (1); −0.75 to −2.0 (2) |
| Micrographs collected (#) | 25,179 | | | | 16,055 | 11,899 (1); 32,907 (2) |
| Pixel size (Å) | 0.654 | | | | 0.654 | 0.654 |
| **Map reconstruction** | | | | | | |
| Image processing package | cryoSPARC | | | | | |
| Total extracted particles | 334,256 | | | 334,256 | 295,185 | 47,449 |
| Final particle count | 184,019 | 184,019 | 17,473 | 76,491 | 230,567 | 24,011 |
| Symmetry imposed | C1 | | C1 | C1 | C1 | C1 |
| **Resolution (Å)** | | | | | | |
| GS-FSC (0.143) | | | | | | |
| *unmasked* | 7.0 | 7.0 | 12 | 8.0 | 9.7 | 15 |
| *loose* | 5.6 | 3.7 | 7.1 | 6.0 | 9.1 | 14 |
| *tight* | 4.4 | 3.5 | 6.6 | 4.5 | 7.7 | 12 |
| 3D-FSC sphericity (out of 1.0) | 0.89 | 0.82 | 0.712 | 0.906 | 0.926 | 0.870 |
| **Model composition and refinement** | | | | | | |
| Number of atoms | 114,335 | 40,604 | | | | |
| Protein residues | 7130 | 2556 | | | | |
| Refinement package | Phenix and Coot | | | | | |
| Map-to-model cross-correlation | | | | | | |
| *masked* | 0.69 | 0.77 | | | | |
| *unmasked* | 0.72 | 0.78 | | no models built | | |
| RMSD bond length (Å) [#>4σ] | 0.003 [0] | 0.003 [0] | | | | |
| RMSD bond angles (o) [#>4σ] | 0.702 [0] | 0.664 [0] | | | | |
| MolProbity score | 0.88 | 0.91 | | | | |
| Clash score | 1.43 | 2.48 | | | | |
| C-beta outliers (%) | 0.0 | 0.0 | | | | |
| Rotamer outliers (%) | 0.03 | 0.0 | | | | |
| Ramachandran favored (%) | 99.9 | 100 | | | | |

Density for the cytoplasmic portions of both FtsH hexamers in the Carboxy-DIBMA map was far stronger than in the higher-resolution map of the DDM-solubilized complex, which may reflect the presence of native lipids or other differences in the protocols used to purify the complexes.

## FtsH•HflK/C enhances lipid scrambling

Unusual membrane curvature often correlates with enhanced rates of polar lipid head group flipping from one side of a bilayer to the other (Devaux et al, 2008; Jarsch et al, 2016). This activity can be assayed by reconstituting proteoliposomes with fluorescent NBD-labeled lipids (Ghanbarpour et al, 2021; Ploier and Menon, 2016), which orient randomly in proteoliposomes, with approximately half of the fluorescent lipids initially in the inner leaflet and half in the outer leaflet. Treatment with dithionite, which does not penetrate liposomes, quenches the fluorescence of the outward-facing NBD groups, thereby reducing the fluorescence to ~50% of the initial level (Fig. 4A, left). If proteins that catalyze lipid scrambling are present in the membrane, dithionite treatment can further reduce fluorescence, as NBD-labeled lipids initially facing the lumen can be translocated to the outside, where they are exposed to dithionite quenching (Fig. 4A, right). Using this assay, we found that FtsH•HflK/C and FtsH catalyze lipid scrambling (Fig. 4B,C). Although this activity did not require ATP hydrolysis by FtsH, we postulate that it may help this ATP-fueled protease to extract membrane proteins from the lipid bilayer during degradation (see "Discussion").

## HflK/C impacts intracellular protein levels

To investigate how HflK/C affects *E. coli* protein levels, we used mass spectrometry to quantify the steady-state abundance of ~2,250 proteins in wild-type cells or mutant cells lacking HflK/C

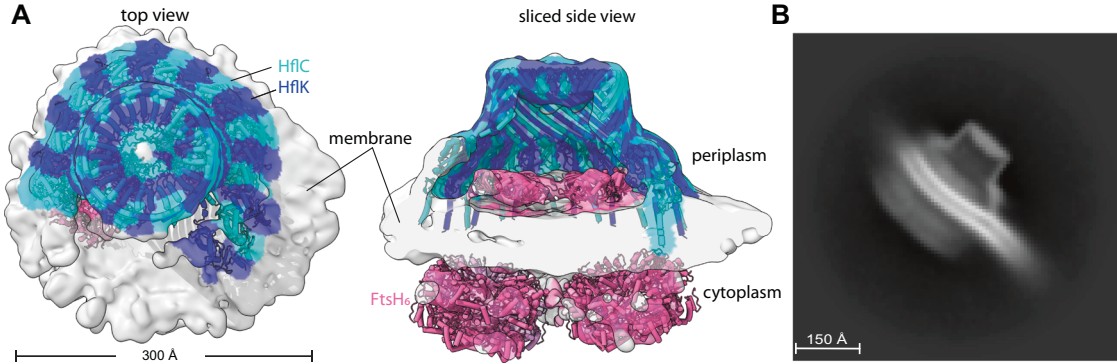

**Figure 3.  Nautilus-like FtsH•HflK/C structure isolated with native lipids via detergent-free solubilization.**

(A) Density map highlighting the nautilus FtsH•HflK/C structure determined using affinity purification following detergent-free solubilization using Carboxy-DIBMA (see "Methods"). Map is color-coded (HflK, blue; HflC cyan; FtsH, pink) using docked atomic model, and displayed from the top (left) and as a sliced side view (right). Key structural elements and scale bar noted. (B) 2D-class average of the Carboxy-DIBMA-solubilized sample highlighting membrane reshaping, as evidenced by the belt-like membrane structure that exhibits dual curvature as it traverses through and extends beyond the particle.

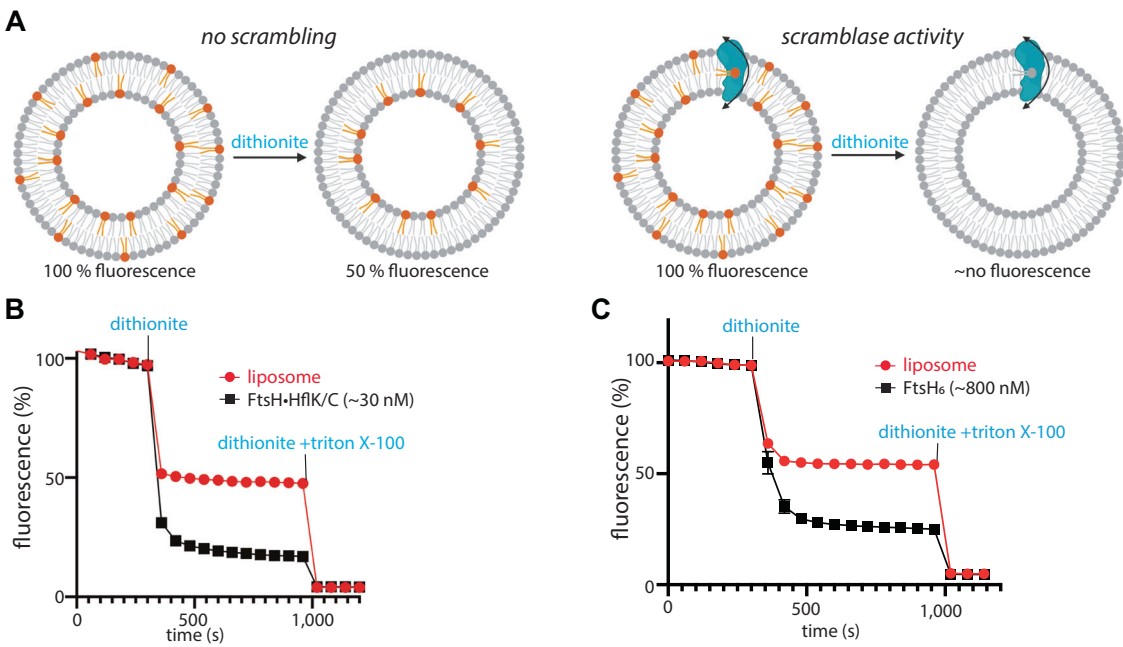

**Figure 4.  FtsH-catalyzed lipid scramblase activity.**

(A) Schematic representation of scramblase activity assay, with expected fluorescence level in the absence (left) or presence (right) of enzymatic lipid scrambling activity. Scramblase activity assay of the FtsH•HflK/C complex (B) or isolated $FtsH_6$ (C) using NBD-PE as a substrate. Note that reported protein concentrations do not consider differences in proteoliposome reconstitution efficiency between samples, and that not all liposomes necessarily contain protein. Control reactions assessing the FtsH-dependence of the measured scramblase activity include: (1) a lack of scramblase activity using detergent-solubilized GlpG membrane protein (Ghanbarpour et al, 2021); and (2) NBD-glucose and BSA-back-extraction assays showing that the measured scramblase activity was not a product of partially disrupted, leaky liposomes (Appendix Fig. S15). Values are means (*n* = 3 technical replicates), with bars marking SEM. Source data are available online for this figure.

($\Delta hflK/C$) (Fig. 5A). Of these proteins, ~88% showed no significant differences, ~8% were more abundant in $\Delta hflK/C$ cells, and ~4% were less abundant in these mutant cells. We considered that FtsH•HflK/C could degrade proteins in the "more abundant" category faster than FtsH alone, and that HflK/C could restrict degradation of proteins in the "less abundant" category. However, other possibilities include indirect effects of *hfl*K/C genetic deletion,

such as an impact on FtsH-driven degradation of proteins that regulate the transcription, translation, or turnover of these differentially abundant proteins. Notably, proteins in the "more abundant" category included the membrane proteins DadA, PspC, SecY, and YlaC (Fig. 5B), which are known to be degraded by FtsH (Arends et al, 2016; Singh and Darwin, 2011; Westphal et al, 2012). To identify proteins that directly interact with FtsH, we affinity-purified FtsH from

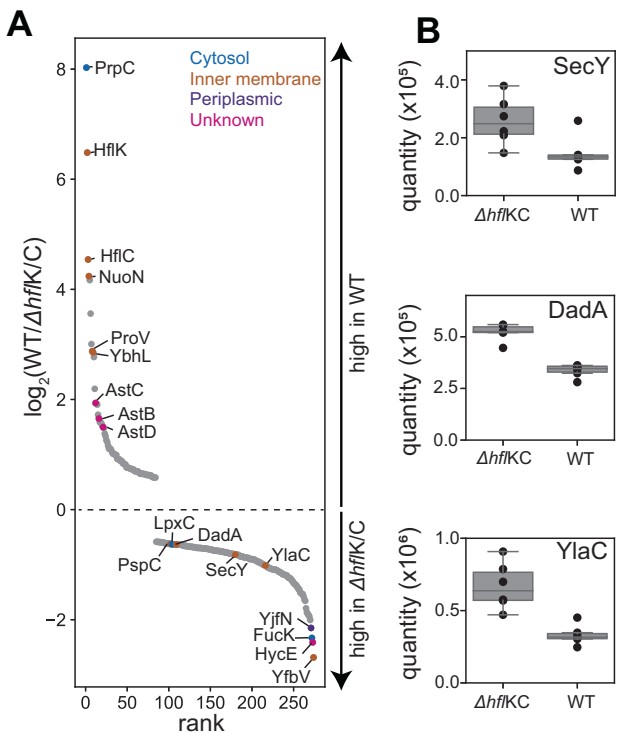

**Figure 5. Steady-state protein abundance measurements in wild-type and Δ*hfl*K/C cells.**

(A) Semi-quantitative proteomic measures of protein abundance in wild-type or Δ*hfl*K/C strains of *E. coli*. Proteins ($n = 274$) exhibiting statistically significant changes (false discovery rate $q < 0.01$, as assessed using an unpaired Student's t-test corrected for multiple hypothesis testing using a Benjamini–Hochberg procedure implemented in Spectronaut) across six replicates are plotted following the measured rank order abundance change between cell types. A subset of proteins are labeled by gene name, and colored by their annotated localization. Among the highlighted Δ*hfl*K/C-enriched proteins are the putative FtsH substrates LpxC, DadA, SecY, and YlaC. (B) Signal intensity (quantified using label-free precursor ion intensities with Spectronaut) compared between wild-type and Δ*hfl*K/C cells. Inner-quartile range (IQR) depicted by box plot, with dots marking abundance measured in each of six replicates, and with whiskers extending 1.5*IQR. Protein levels changes were statistically significant, with multiple hypothesis testing corrected $q$ values of $7.9 \times 10^{-5}$ (SecY), $3.6 \times 10^{-11}$ (DadA), and $8.2 \times 10^{-7}$ (YlaC).

wild-type or Δ*hfl*K/C cells and analyzed the eluate by mass spectrometry. Notably, some of the differentially abundant proteins identified above, including DadA and AstC, also exhibited HflK/C-dependent association with FtsH (Dataset EV1).

## Discussion

In *E. coli*, the SPFH-family HflK and HflC proteins interact with the AAA FtsH protease to form a large holoenzyme (Kihara et al, 1996; Saikawa et al, 2004). The first cryo-EM structures, which were solved after protein overproduction using C4-symmetry in the reconstruction process, showed HflK/C subunits forming a tight symmetric cage around four FtsH hexamers (Ma et al, 2022; Qiao et al, 2022). In these structures, the transmembrane helices of HflK/C completely surround the internal FtsH hexamers and would appear to insulate FtsH from interactions with potential membrane-embedded substrates. By contrast, following

purification of FtsH•HflK/C without overexpression, the nautilus-like cryo-EM structures presented here, which were determined without imposed symmetry, revealed an HflK/C chamber containing one or two FtsH hexamers and an opening that could allow membrane proteins to enter and be engaged by FtsH. In the study by Ma et al, 3D classes with a nautilus-like structure and fewer than four FtsH hexamers were observed but were interpreted as ruptured states resulting from purification artifacts. Additionally, in the study by Qiao et al, factors beyond the overexpression of HflK/C and FtsH, including such use of chemical crosslinking or mutations in FtsH may have contributed to the closed-cage structures observed.

Although it is possible that a population of cellular FtsH•HflK/C corresponds to the symmetric caged structure, measured cellular protein stoichiometries in *E. coli* (Li et al, 2014) preclude such structures being uniformly present. Specifically, the abundances of FtsH, HflK, and HflC in several growth conditions are only sufficient for an average of ~2.3 FtsH hexamers for every 24 subunits of HflK/C (Li et al, 2014). Moreover, a model in which cells contain approximately equal amounts of free FtsH hexamers and sets of four FtsH hexamers tightly caged by HflK/C is inconsistent with experiments showing that the vast majority of FtsH co-sediments in a large complex with HflK/C, with very few free FtsH hexamers observed (Saikawa et al, 2004). Our affinity isolation experiments were also consistent with this observation, as the stoichiometry of the components in our purified FtsH•HflK/C sample (Appendix Fig. S2) was similar to that in cells. Thus, we posit that the nautilus-like FtsH•HflK/C structures described here likely account for a major fraction of holoenzyme complexes in the cell. Notably, most bacteria that encode FtsH also encode HflK and HflC homologs, and mitochondrial relatives of FtsH associate with SPFH-family proteins (Fu and MacKinnon, 2024; Wessel et al, 2023), raising the possibility that such super-complexes are present in a variety of organisms.

Structurally, one would expect the previously reported symmetric HflK/C enclosure to prevent FtsH degradation of membrane-embedded substrates, whereas the nautilus-like structure with its open portal could enhance degradation of some membrane proteins and slow degradation of others. For example, interactions between membrane-embedded proteins and structural elements of HflK/C within the nautilus chamber could increase substrate residence times, thereby making the engagement of degron sequences by the cytoplasmic proteolytic machinery of FtsH more probable. Supporting this idea, our proteomic experiments showed that deletion of *hfl*K/C increased the abundance of several membrane proteins that were previously shown to be substrates of FtsH (Arends et al, 2016; Rei Liao and van Wijk, 2019; Westphal et al, 2012) (Fig. 5A,B). Additional membrane and soluble proteins also had different steady-state levels in wild-type and Δ*hfl*K/C cells, but whether these proteins are direct substrates or their levels are affected by another FtsH•HflK/C substrate remains to be determined. Conversely, the entryway into the HflK/C chamber, which spans ~70 Å, could prevent the entry and degradation of sufficiently large membrane-protein complexes, with the HflK/C nautilus negatively regulating FtsH activity. Finally, HflK/C appears to influence the associated lipid domain and membrane localization of FtsH (Wessel et al, 2023), which could be important in determining the local pool of membrane-embedded substrates subject to FtsH-mediated degradation.

We observed reverse-membrane curvature in our FtsH•HflK/C complexes, which was most evident in the detergent-free nanodisc samples (Fig. 3). Reverse-membrane curvature, coupled with a potentially different lipid composition within the FtsH•HflK/C domain (Browman et al, 2007; Wessel et al, 2023), might facilitate the

recruitment of certain membrane-embedded substrates or adaptors into the FtsH•HflK/C complex. Indeed, in eukaryotic cells, altered membrane curvature is often seen within lipid rafts, which can act as cellular reaction centers within the plasma membrane (Sadeghi et al, 2014). FtsH and FtsH•HflK/C each catalyze the rate at which lipids change orientation within membrane bilayers. A simple way in which this could occur is if FtsH and the complex were to thin parts of the associated membrane, as this would reduce the energetic penalty for passage of the head group through the hydrophobic interior of the membrane. In mitochondria, FtsH homologs regulate lipid metabolism (Chiba et al, 2006; Löser et al, 2021), which may depend on enhanced lipid flipping and downstream processes. Membrane thinning could also assist in the degradation of membrane-embedded substrates by FtsH. Here, we envision that the free-energy cost of extracting a membrane protein from the bilayer during degradation would be reduced if the membrane were thinner, an idea that has been advanced to explain the CDC48-dependent degradation of ERAD-L substrates in eukaryotes (Wu et al, 2020). Taken together, the unique structural features of this super-complex, including the asymmetric nautilus-like assembly and its impact on the local membrane curvature may support FtsH's critical role in maintaining proteostasis at the membrane.

# Methods

### Reagents and tools table

| Reagent/resource | Reference or source | Identifier or catalog number |
| --- | --- | --- |
| **Experimental models** | | |
| *E. coli* BL21 FtsH-FLAG (C-terminal) | This work following: Reisch CR and Prather KLJ (2017) | Strain AG1_FR1_B1_A1 |
| BL21 FtsH-FLAG (C-terminal) ΔhflK/C | This work following: Reisch CR and Prather KLJ (2017) | Strain AG2_Fr1_B1_A2 |
| **Recombinant DNA** | | |
| BL21_C-FLAG construct | Integrated DNA Technologies, Inc. | Sequence provided as source data file BL21_C-flag.dna |
| pKDsgRNA-p15a | Addgene | Plasmid #62656 |
| pCas9-CR4 | Addgene | Plasmid #62655 |
| GTGAAAGCACTGATTGAG CGTAACTATAATCGTGCGCGT CAGCTTCTGACCGACAATATG GATATTCTGCATGCGATGAAA GATGCTCTCATGAAATATGAG ACTATCGACGCACCGCAGATT GATGACCTGATGGCACGTCGC GATGTACGTCCGCCAGCGGGC TGGGAAGAACCAGGCGCTTCT AACAATTCTGGCGACAATGGT AGTCCAAAGGCTCCTCGTCC GGTTGATGAACCGCGTACGCC GAACCCGGGTAACACGA **TGTCAGAGCAGTTAGGCGAC** AAGAGCGGGTCAGGA GACTATAAGGACGACGAC GACAAGTAA | Integrated DNA Technologies, Inc. | Geneblock sequence containing FtsH Flag Tag (shown underline) and protospacer (shown in bold) |

| Reagent/resource | Reference or source | Identifier or catalog number |
| --- | --- | --- |
| **Antibodies** | | |
| NA | NA | NA |
| **Oligonucleotides and other sequence-based reagents** | | |
| Recombineering oligos | Integrated DNA Technologies, Inc. | NA |
| **Chemicals, enzymes, and other reagents** | | |
| Pyruvate Kinase/Lactic Dehydrogenase enzymes from rabbit muscle | Sigma Aldrich, Inc. | P0294 |
| NADH, grade II | Roche, Inc. | CAS# 606688 |
| Phosphoenolpyruvate | Roche, Inc. | 10108294001 |
| Carboxy-DIBMA | Dr. Sandro Keller's lab University of Graz | Described in this work |
| Graphene-supported grids | Grassetti et al, 2023 | 200-mesh Quantifoil 2/1 copper grids, pre-treated with UV/ozone for 7 min using a Bioforce UV/ozone cleaner |
| 1-myristoyl-2-{6-[(7-nitro-2-1,3-benzoxadiazol-4-yl)amino] hexanoyl}-sn-glycero-3-phosphocholine | Avanti Inc. | 810122 |
| POPC | Avanti Inc. | 850457 |
| POPE | Avanti Inc. | 850757 |
| NBD-PE | Avanti Inc. | 810153C |
| DDM | Anatrace Inc. | D310 |
| GDN | Anatrace Inc. | GDN101 |
| Bio-Beads SM-2 Adsorbents #1523920 | Bio-Rad Inc. | #1523920 |
| S-Trap Micro Columns | Protifi inc. | C02-micro-80 |
| **Software** | | |
| CryoSPARC | Punjani et al, 2017 | V3.0; v4.0 |
| ChimeraX | Pettersen et al, 2021 | V1.2 |
| Coot | Casanal et al, 2020 | V0.9.4 |
| Phenix | Liebschner et al, 2019 | V1.14 |
| cryoDRGN | Zhong et al, 2021 | V0.3.5 |
| Warp | Tegunov and Cramer, 2019 | V1.0.9 |
| IMOD | Mastronarde and Held, 2017 | V4.11.12 |
| RELION | Zivanov et al, 2018 | V3.1.3 |
| M | Tegunov et al, 2021 | V1.0.9 |
| tomoDRGN | Powell and Davis, 2024 | V0.2.2 |
| Spectronaut | Biognosys AG | V15 |

| Reagent/resource | Reference or source | Identifier or catalog number |
|---|---|---|
| **Other** | | |
| Titan Krios G3i – Gatan BioQuantum K3 | Thermo Fisher and Gatan | n/a |
| Q-exactive HF-X | Thermo Fisher | n/a |

## FtsH•HflK/C purification

A C-terminal FLAG-tagged variant of FtsH was introduced into *E. coli* BL21(DE3) at the endogenous FtsH locus using the no-SCAR (Scarless Cas9 Assisted Recombineering) system (Reisch and Prather, 2017). Briefly, the spacer sequence GTCGCCTAACTGCTCTGACA targeting the C-terminus of FtsH was cloned into the temperature-sensitive pKDsg-XXX plasmid, and introduced into BL21(DE3) cells containing the pCas9cr4 plasmid. λ-Red recombineering (Datsenko and Wanner, 2000) was induced, followed by transformation with a 500 bp double-stranded gene block (IDT) containing a mutation in the PAM sequence that appended the sequence coding for "SGSGDYKDDDK" to the C-terminus of FtsH. The chromosomal mutation was selected for by inducing Cas9 and subsequently verified by amplifying the region using PCR and DNA sequencing. BL21(DE3) cells expressing this C-terminal FLAG-tagged FtsH were grown overnight in 50 mL of Luria broth (LB), transferred to a 1:1 LB:2xYT medium mixture, and grown for an additional 20 h to an $A_{600}$ of ~5.5. Cells were harvested by centrifugation at 4500 rpm for 30 min and resuspended in buffer A [20 mM HEPES (pH 7.5), 400 mM NaCl, 100 mM KCl, 10% glycerol, 1 mM DTT] before flash freezing in liquid $N_2$ for storage at $-80\,°C$.

For cell lysis, resuspended cells were sonicated (QSonica) on ice using 30% power with cycles of 10 s on and 30 s off for a total duration of 4 min. Following sonication, cells were centrifuged at $12,100 \times g$ for 20 min in a JL25.50 rotor. The supernatant was isolated and then centrifuged at $105,000 \times g$ for 1 h using a 45 Ti rotor. The pelleted membrane was resuspended in 5 mL of buffer A and disrupted using a Dounce homogenizer. Detergent *n*-dodecyl-β-D-maltoside (DDM) was added to a final concentration of 1%, followed by platform rotation at 4 °C for 1 h. The mixture was centrifuged for 20 min at 35,000 rpm using a TLA 100.3 rotor, incubated with anti-Flag M2 resin overnight, washed with a buffer containing 0.03% DDM, and finally eluted with 0.25 mg/mL Flag peptide (APEXBio) according to the manufacturer's directions. The same protocol was used for glyco-diosgenin (GDN) sample preparation, except 2% GDN was used for membrane solubilization and 0.03% GDN was employed for the wash and elution steps.

The overexpressed HflK/C sample was prepared using the petDUET plasmid as previously reported (Ma et al, 2022). Briefly, HflC followed by a C-terminal Flag-tag was cloned into the first multicloning site, and HflK with an N-terminal octa-histidine tag was cloned into the second site. BL21 (DE3) cells harboring the plasmid were grown at 37 °C from an overnight cell culture to an OD of ~1. Expression was induced by the addition of 0.2 mM IPTG and continued overnight at 18 °C. Purification was performed similarly to the procedure described for the endogenously expressed FtsH HflK/C solubilized in DDM.

## Carboxy-DIBMA extraction

Carboxy-DIBMA was synthesized analogously to Sulfo-DIBMA as described (Glueck et al, 2022). Membranes, isolated as described above, were resuspended in buffer B [50 mM Tris-HCl (pH 8.0), 100 mM NaCl, 5 mM $MgSO_4$, 100 μM $ZnCl_2$, and 3 mM β-mercaptoethanol], to a final concentration of ~35 mg/mL. To this suspension, 22 mg/mL of Carboxy-DIBMA solution and 1 mM ATP were added, bringing the total volume to 5 mL. The mixture was gently rotated overnight at 4 °C to facilitate solubilization. Subsequently, samples were centrifuged at $30,000 \times g$, yielding a clear supernatant. This supernatant was incubated with anti-FLAG M2 magnetic beads for 2 h at 4 °C with gentle rotation. After incubation, the sample was centrifuged at $300 \times g$, the supernatant was carefully removed with a magnet, and beads were washed with 15 mL of buffer B. Finally, the sample was eluted using 0.5 mg/mL FLAG peptide. The eluted sample was concentrated, analyzed, and quantified after SDS-PAGE, using bovine serum albumin as a standard.

## Cryo-EM sample preparation

DDM- or GDN-solubilized samples were prepared by applying 2.5 μL of ~0.75 mg/mL of FtsH•HflK/C complex onto homemade graphene-supported 200-mesh Quantifoil 2/1 copper grids (Grassetti et al, 2023), pre-treated with UV/ozone for 7 min using a Bioforce UV/ozone cleaner. Sample-loaded grids were blotted for 4 s with a blot force of +4 at 6 °C and 95% relative humidity using a FEI Vitrobot Mark IV instrument. The same concentration of the overexpressed HflK/C sample was used for grid preparation. However, in this case, 2-nm carbon-supported 200-mesh Quantifoil 2/1 copper grids, which had been glow-discharged for 20 s in an easiGlow glow discharger (Pelco) at 15 mA, were utilized. For the Carboxy-DIBMA-extracted sample, a concentration of ~0.25 mg/mL of the FtsH•HflK/C complex was applied to 2-nm carbon-supported 200-mesh Quantifoil 2/1 copper grids, also glow-discharged for 20 s in an easiGlow glow discharger (Pelco) at 15 mA, and the sample was vitrified as above.

## Cryo-EM data collection

For the DDM-solubilized FtsH•HflK/C complex, 25,179 movies were collected with EPU using aberration-free image shift (AFIS) and hole-clustering method on a Titan Krios G3i with an acceleration voltage of 300 kV and magnification of 130,000×, detected in super-resolution mode on a Gatan K3 detector for an effective pixel size of 0.654 Å (binned by 2). Movies were collected as 40 frames with a defocus range from $-0.05$ to $-1.75$ μm and a total exposure per specimen of 46.1 e-/Å². For the DDM-solubilized and overexpressed HflK/C complex, movies were collected as 40 frames with a defocus range from $-0.5$ to $-1.75$ μm and a total exposure per specimen of 47.2 e-/Å² using AFIS hole clustering. For the Carboxy-DIBMA-extracted sample, movies (11,989) were collected as 40 frames with a defocus range from $-0.30$ to $-1.75$ μm and a total exposure per specimen of 44.6 e-/Å² using AFIS hole clustering. An additional data collection session on this sample produced 32,907 movies, collected with a defocus range from $-0.75$ to $-2.0$ μm and a total exposure per specimen of 44.4 e-/Å² using AFIS hole clustering.

## Cryo-EM pre-processing and particle picking

For the DDM-solubilized FtsH•HflK/C complex, data processing was performed using cryoSPARC (Punjani et al, 2017) (v4.0) and default parameters unless noted (see Appendix Fig. S3). Raw movies (25,179) were pre-processed using "Patch motion correction", and "Patch CTF estimation". Particles (73,171) were picked using the "Blob-picker" tool applied to 1000 micrographs selected at random. Particles were extracted (box size 640 px, Fourier-cropped to 360 px) and classified using the '2D classification' utility. The entire set of micrographs were then picked with the "Template picker" tool, using a single class from 2D classification. Particles were extracted (box size 640 px, Fourier-cropped to 360 px) and subjected to three rounds of 2D classification, resulting in the selection of 334,256 particles as a preliminary stack.

For the DDM-solubilized overexpressed HflK/C complex, data processing was performed using cryoSPARC (v4.0) and default parameters unless noted (see Appendix Fig. S11). Raw movies (16,055) were pre-processed and particles picked as described above, except that the extraction-box-size was 900 px (Fourier-cropped to 300 px). After four rounds of 2D classification, 295,185 particles were selected as a preliminary stack.

For the FtsH•HflK/C complex extracted with carboxy-DIBMA, data processing was carried out using cryoSPARC (v4.0) with default settings unless noted (see Appendix Fig. S13). Raw movies were collected over two sessions. Data from these sessions were pre-processed individually using 'Patch motion correction' and 'Patch CTF estimation'. Particles were separately extracted (box size of 1000 px, Fourier-cropped to 360 px). After several rounds of 2D classification of particles from the isolated datasets, the particles were pooled and subjected to two additional rounds of 2D classification, resulting in the final particle stack of 47,449 particles.

## Ab initio reconstruction, global refinement, and model building

For the FtsH•HflK/C complex solubilized in DDM, "Ab-initio reconstruction" using two classes was performed, with a resulting single class model and particle stack (184,019 particles) selected for subsequent "Homogenous refinement" followed by "Non-uniform refinement". A mask was then created in ChimeraX (Pettersen et al, 2021) that only included the periplasmic portion of HflK/C and FtsH, excluding the cytoplasmic domain of FtsH, which exhibited substantial flexibility. Using this mask, 'Local refinement' was performed, leading to a ~4.4 Å GS-FSC resolution map we termed "map a". To separate single vs. double FtsH hexamers within the HflK/C assembly, a mask was generated that encompassed the FtsH periplasmic domain with a lower occupancy. Using this mask, '3D classification' (20 classes) was performed. Based on visual inspection, particles assigned to maps bearing one FtsH hexamer were pooled and subjected to "Local refinement" using the original mask, producing "map I". A parallel operation was performed on particles assigned to maps bearing two hexamers, resulting in "map II".

To generate a higher-resolution map for the "hat" portion of the complex a signal subtraction and local refinement strategy was employed (see Appendix Fig. S7). Briefly, two masks were generated using map a - one encompassing on the hat portion and another containing the remainder of map a, which was used for signal subtraction after first locally aligning the particles using the

non-hat mask using the "Local refinement" utility. Next, the signal-subtracted particles and the mask around the hat portion were used to generate an initial volume using the "Homogeneous reconstruct only" utility followed by a "Local refinement" using the same mask. After application of the "Global CTF refinement" followed by "Local CTF refinement", the final map reached ~3.5 Å GS-FSC resolution.

For the overexpressed HflK/C complex solubilized in DDM, "Ab initio reconstruction" using three classes was performed, and one class was selected (106,858 particles). "Homogeneous refinement" followed by "Non-uniform refinement" were then employed, producing a map that was used as an initial model (after low-pass filtering) in a final round of "Non-uniform refinement" followed by "Local refinement" with the original particle stack (230,567 particles). After conducting both global and local CTF refinements, the final map achieved a GS-FSC resolution of ~9.8 Å.

For the FtsH•HflK/C complex extracted with Carboxy-DIBMA, 'Ab initio reconstruction' was carried out on the 47,449 particles using two classes. Of these, 24,011 particles were selected for "Non-uniform refinement", followed by global and local CTF refinement, and an additional round of "Local refinement".

Model building was performed using a combination of ChimeraX-1.3 (Pettersen et al, 2021), Coot (0.9.4) (Casanal et al, 2020), and Phenix (1.14) (Liebschner et al, 2019). The PDB ID 7WI3 served as the initial model, but it aligned well only with the "hat" portion of the HflK/C assembly density. Therefore, the individual HflK and HflC subunits were docked separately into the map. Final model building was performed iteratively using Phenix, with manual fitting in Coot to enhance the fit between the map and model. Local resolution was estimated by cryoSPARC implementation of monoRes (Vilas et al, 2018).

## Tilt-series acquisition

The GDN-solubilized FtsH•HflK/C sample was used for tomographic data collection. A total of 18 tilt series were acquired on a Titan Krios G3i microscope operating at 300 kV. Movies were collected at a nominal magnification of 64,000× on a BioQuantum K3 detector operating in super-resolution mode, and were pre-binned at collection time to a physical pixel size of 1.36 Å, with 6 frames per movie. Tilt series were collected following a dose-symmetric scheme from −58° to 58° in 2° increments, beginning from 0°, and each tilt series was collected at a nominal defocus stepped between −3 μm and −5 μm. The total electron dose per tilt series was 117.2 e$^-$/Å$^2$. The energy-filter slit width was 20 eV during acquisition, and the zero-loss peak was refined between each tilt movie acquisition.

## Tomogram reconstruction and sub-tomogram averaging

Tomographic data processing generally followed Powell et al, (Powell et al, 2025), with the following modifications. Tilt movies were aligned, and initial CTF parameters estimated in Warp (v1.0.9) (Tegunov and Cramer, 2019). Tilt-series stacks were exported and aligned in IMOD (v4.11.12) using patch tracking (Mastronarde and Held, 2017). Tilt series were re-imported into Warp with alignment parameters and CTF parameters were fit using each full-tilt series. Tomograms were reconstructed and deconvolved in Warp using a pixel size of 10 Å/px. All particles

were manually picked using Cube ($n = 1578$ particles). Particles were exported from Warp with pixel size of 8 Å/px and box size of 64 px$^3$. RELION (v3.1.3) (Zivanov et al, 2018) was used for initial model generation and subsequent particle refinement with C1 symmetry, reaching an unmasked resolution of 24.4 Å. Refined particles were imported to M (v1.0.9) (Tegunov et al, 2021) and subjected to iterative refinement of image warp, volume warp, particle pose, and stage angle parameters, ultimately reaching a masked resolution of 17.9 Å. Particles were re-extracted from M with pixel size of 8 Å/px and a box size of 64 px$^3$, re-refined in RELION, re-imported to M, and re-refined in M, reaching a final resolution of 17.3 Å. Copies of this final reconstruction were mapped back in the source tomogram and visualized with ChimeraX using the subtomo2chimerax functionality implemented in tomoDRGN (Powell and Davis, 2024).

## Mass spectrometry

For steady-state protein comparisons between wild-type *and Δhfl*K/C *E. coli* strains, cells were grown in LB broth and harvested via centrifugation at 4 °C at an A$_{600}$ of ~1. Cell pellets were frozen at −80 °C for future use. Thawed cell pellets were washed with buffer MS1 [20 mM Tris-HCl (pH 7.5), 100 mM NH$_4$Cl, 10 mM MgCl$_2$, 0.5 mM EDTA, and 6 mM β-mercaptoethanol], followed by the addition of SDS (10% final concentration) and 0.25 U/µL benzonase (EMD Millipore). After incubation for 10 min at room temperature, lysates were cleared via centrifugation at $21,000 \times g$ for 8 min. DTT (final concentration of 20 mM) was introduced to the lysate, followed by heating at 65 °C to reduce disulfide bonds. After cooling to 30 °C, iodoacetamide (final concentration of 40 mM) was added for 30 mins to alkylate-free cysteines. Subsequent acidification, column preparation, and sample binding to S-Trap™ micro (≤ 100 µg) spin columns and column washing followed the manufacturer's protocol (ProtiFi). A trypsin solution (Promega Inc.) was combined with 200 µL of digestion buffer and applied to the spin columns containing bound proteins followed by a 1 h incubation at 42 °C. Peptides were eluted using a sequential addition of 50 mM TAEB and 0.2% (v/v) formic acid. Further elution with acetonitrile-formic acid mix (35 µL of 50% acetonitrile, 0.2% formic acid) helped maximize the recovery of hydrophobic peptides. Finally, peptides were dried in a speed-vac and resuspended in buffer MS2 [4% acetonitrile, 0.1% formic acid] for subsequent mass spectrometry analysis.

Peptides (~1.5 µg) were injected into an Ultimate 3000 UHPLC system (Thermo Scientific) comprising a PepMap100 C18 (75 µm ID × 2 cm, particle size 3 µm, pore size 100 Å; Thermo Fisher Scientific) precolumn upstream of a PepMap C18 (75 µm ID × 2 cm, particle size 3 µm, pore size 100 Å; Thermo Fisher Scientific) analytical column. Sample loading proceeded at 300 nL/min in 98% buffer LCMS_A [0.1% formic acid in water], 2% LCMS_B [0.1% formic acid in acetonitrile]. After washing the precolumn with 20 column volumes of LCMS_A, peptides were resolved using a 120-min linear gradient from 4% to 30% LCMS_B. Peptides were ionized by nanospray electrospray ionization and analyzed on a Q Exactive HF-X mass spectrometer (Thermo Scientific). For data-dependent acquisitions, MS1 full scans (350–1400 $m/z$ range) were collected at a resolution of 60,000 (AGC 3x10$^6$, 50 ms maximum injection time). HCD fragmentation of the 12 most abundant precursor ions was performed at 25% NCE. The fragment analysis

(MS2) was performed with a resolution of 15,000 (AGC $1 \times 10^5$, 100 ms maximum injection time, 2.2 $m/z$ isolation window). For data-independent acquisitions, MS1 full scans (350–1400 $m/z$ range) were collected at a resolution of 120,000 (AGC $3 \times 10^6$, 50 ms maximum injection time). Fragment analysis (MS2) was subdivided into 25 DIA isolation windows of varying widths (see Appendix Table S1) using a resolution of 30,000 (AGC $1 \times 10^6$, 70 ms maximum injection time). HCD fragmentation was performed at 25% NCE. The resulting raw DIA files were analyzed using Spectronaut (v15), with a search database containing the *E. coli* Uniprot database concatenated to a database containing common contaminants. In silico digests were performed using Trypsin/P and LysC. The false discovery rate was set at 1% at both the peptide and protein levels. Normalization of all replicates and generation of the volcano plot were performed using Spectronaut.

For the preparation of samples for mass spectrometric analysis of the pull-down of FtsH from wild-type and *Δhfl*K/C strains, the same purification protocol for the FtsH•HflK/C complex described above was followed, using GDN detergent to retain weak interactions between FtsH and its substrates. After running an SDS-PAGE gel of the FLAG resin eluate from WT and *Δhfl*K/C cells, the entire gel lane for each condition was excised and sent to the proteomic core facility at the Koch Institute for Cancer Research (MIT) for data analysis. There, samples were analyzed via DDA on a Q Exactive HF-X mass spectrometer coupled to a Ultimate 3000 UHPLC. Co-purifying proteins in each condition were quantified using spectral counts, normalized to that of FtsH in each sample.

## Biochemical assays

Expression and purification of FtsH-bearing samples followed the protocols listed above, using the following buffer substitutions. Buffer A was substituted with buffer BA1 [50 mM Tris-HCl (pH 8.0), 1 mM MgCl$_2$, 100 mM KCl, 10% glycerol, and 3 mM β-mercaptoethanol] during cell lysis, and with buffer BA2 [50 mM Tris-HCl (pH 8.0), 100 mM NaCl, 5 mM MgCl$_2$, 100 µM ZnCl$_2$, 10% glycerol, 3 mM β-mercaptoethanol] when resuspending the pelleted membranes. In addition, prior to elution, the resin was washed using buffer BA3 [50 mM Tris-HCl (pH 8.0), 400 mM NaCl, 5 mM MgCl$_2$, 100 µM ZnCl$_2$, 0.03% DDM and 10% glycerol]. The protein was eluted with 0.25 mg/mL FLAG peptide (APEXBio) in buffer BA5 [50 mM Tris-HCl (pH 8.0), 100 mM NaCl, 5 mM MgCl$_2$, 100 µM ZnCl$_2$, 10% glycerol, and 0.03% DDM] according to the manufacturer's directions.

The λ-CII substrate was produced using HisTag-SUMO-λ-CII in the pET22b plasmid. The BL21(DE3) cells harboring the plasmid were grown to an OD$_{600}$ of ~0.8 at 37 °C. Protein expression was induced by the addition of 0.5 mM of IPTG, and the culture was grown overnight at 18 °C. Cells were harvested by centrifuging at 4000 rpm for 25 min at 4 °C. The cell pellet was resuspended in buffer BA6 [20 mM HEPES (pH 8.0), 400 mM NaCl, 100 mM KCl, and 10% glycerol], and stored at −80 °C.

For purification, the cells were thawed and incubated with 1 mM DTT and protease inhibitor cocktail prior to lysis using 30% power with cycles of 10 s on and 30 s off for a total duration of 3.5 min. The sample was centrifuged at 4500 rpm for 30 min in a JA25.50 rotor to remove cell debris. The supernatant was transferred to a new tube and incubated with pre-washed Ni-NTA resin for 30 min at 4 °C in buffer BA6 supplemented with 5 mM imidazole. The

resin was washed with 200 mL of buffer BA6 supplemented with 15 mM imidazole to remove impurities. The resin was then incubated with the Sumo Protease overnight at 4 °C in buffer BA6 containing 5 mM imidazole. The next day, the cleaved protein was collected as HisTag-SUMO remained bound to the resin. The cleaved protein was concentrated using a 10-kDa cutoff centrifugal unit and loaded onto a S200 (16/600) gel filtration column. Fractions were collected, analyzed using SDS-PAGE, concentrated, aliquoted, and flash-frozen in liquid nitrogen for future use.

Degradation assays were performed at 37 °C using 5 μM λ-CII, ~0.3 μM of FtsH•HflK/C complex in buffer BA7 [25 mM HEPES (pH 7.5), 150 mM NaCl, 5 mM MgCl₂, 3 mM β-mercaptoethanol,100 μM ZnCl₂, 5 mM ATP, 0.009% DDM]. The concentrations of FtsH/HflK/C were calculated using a BSA standard after running SDS - PAGE. All components except λ-CII were initially mixed at 37 °C for 5 min. λ-CII, which had also been preincubated at 37 °C, was then added to initiate degradation, which was monitored by SDS-PAGE gel analysis.

ATP hydrolysis was measured using a coupled enzymatic reaction (Norby, 1988) in which NADH oxidation to NAD+ reduces absorbance at 340 nm ($\Delta \varepsilon = 6.22\,\mathrm{mM^{-1}\,cm^{-1}}$) using a SpectraMax M5 plate reader and a 384-well assay plate (Corning, 3575). A 20X stock ATPase reaction mix contained 20 μL of pyruvate kinase and lactic dehydrogenase from rabbit muscle (P0294, Sigma Aldrich), 10 μL of 200 mM NADH, grade II (CAS# 606688), 15 μL of 1 M phosphoenolpyruvate (Sigma Aldrich, 10108294001) in 25 mM HEPES-KOH (pH 7.6), and 25 μL of 200 mM ATP. For assays, FtsH₆ (0.5 μM) was present in 10 μL of buffer BA7. After 5 min of equilibration at 37 °C, the ATPase assay was initiated by the addition of an equal volume of 1X ATPase reaction mix.

## Lipid preparation

All lipids, including POPC (#850457C), POPE (#850757C), and NBD-PE (#810153C), were purchased from Avanti lipids, and liposomes were prepared as described (Ghanbarpour et al, 2021). Briefly, a mixture of POPC (90% of total lipid by weight), POPE (9.5%), and NBD-lipid (0.5%, either NBD-PE or NBD-PC) solubilized in chloroform was mixed and dried under a nitrogen stream followed by vacuum to form a lipid film. The dry lipid film was subsequently resuspended in buffer LP1 [50 mM HEPES (pH 7.6), 200 mM NaCl] to create a 10.5 mM lipid stock, which was incubated at 37 °C for 10 min, followed by resuspension and ten freeze-thaw cycles. Finally, the lipid mixture underwent 30 cycles of extrusion through a 400-nm polycarbonate filter to complete the preparation.

## Proteoliposome preparation

Liposomes, with a lipid concentration of 5.25 mM in a total volume of 250 μL, were subjected to destabilization by adding Triton X-100. The concentration of Triton X-100 used was determined based on a prior swelling assay (Ghanbarpour et al, 2021; Marek and Gunther-Pomorski, 2016; Ploier and Menon, 2016). This destabilization process was carried out at room temperature for 2–3 h. Proteins solubilized in detergent were then introduced to the destabilized liposomes, and the mixture was rotated gently for 1 h to ensure mixing. Removal of detergent was achieved in three stages using pre-washed biobeads (Bio-Rad #1523920). Initially, 20 mg of biobeads were added to the mixture, which was then incubated at room temperature for 1 h. Subsequently, 20 mg of additional biobeads were added, and the mixture was allowed to rotate

at room temperature for 2 h. In the final step, the mixture was transferred into a new tube containing 40 mg of fresh biobeads and rotated at 4 °C overnight to complete detergent removal, prior to removing biobeads by pipetting.

## Scramblase assay

Lipid scramblase activity was monitored at 30 °C using 96-well plates containing 100 μL of either liposomes or proteoliposomes. The assay tracked the decrease in NBD fluorescence (excitation 460 nm; emission 538 nm) upon the introduction of dithionite to a final concentration of 5 mM using a BioTek Synergy H1 Hybrid Multi-Mode Reader. Subsequently, an additional dose of dithionite (5 mM) along with Triton X-100 (0.5%) was added to dissolve liposomes and ensure NBD reduction.

To test for proteolipid disruption in these experiments, a 1-myristoyl-2-C6-NBD-PE bovine serum albumin (BSA) back-extraction assay (Ghanbarpour et al, 2021; Marek and Gunther-Pomorski, 2016; Ploier and Menon, 2016) was carried out adding 3 mg/mL BSA to liposomes or proteoliposomes instead of dithionite. Like dithionite, BSA interacts exclusively with NBD-PE from the outer leaflet of liposomes, but BSA quenches fluorescence less effectively than dithionite. The protocol used above was used for the NBD-glucose assay, except NBD-lipids were excluded from liposomes or proteoliposomes and NBD-glucose at a concentration of 12.6 μM was introduced during the liposome-destabilization phase. Glucose leakage was assessed after 2 days of dialysis against buffer free of NBD-glucose.

## Data availability

The raw electron micrographs have been deposited at the Electron Microscopy Public Image Archive (https://www.ebi.ac.uk/empiar/) with accession numbers 12471 and 12472. Density maps and atomic models are available through the EMDB (https://www.ebi.ac.uk/emdb/) with accession numbers 46056, 46057, 46058, 46059, 46061, 46062, and the PDB (https://www.rcsb.org/) with accession numbers 9CZ1 and 9CZ2. Raw mass spectrometry data is available at MASSive (http://massive.ucsd.edu/ProteoSAFe/static/massive.jsp) with the identifier MSV000096162. Correspondence for material and data in the manuscript should be addressed to jhdavis@mit.edu or alirezag@wustl.edu.

The source data of this paper are collected in the following database record: biostudies:S-SCDT-10_1038-S44318-025-00408-1.

## Peer review information

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

## Acknowledgements

This work was supported by NIH grants R01-GM144542 and R35-GM141517, The Smith Family Odyssey Award, and NSF-CAREER grant 2046778. Samples were prepared at the Automated Cryogenic Electron Microscopy Facility in MIT.nano and screened on a Talos Arctica microscope, which was a gift from the Arnold and Mabel Beckman Foundation. We thank Dr. Andrew Grassetti for the gift of graphene-coated grids.

## Author contributions

**Alireza Ghanbarpour**: Conceptualization; Supervision; Investigation; Visualization; Methodology; Writing—original draft; Writing—review and editing. **Bertina Telusma**: Software; Investigation; Visualization; Methodology; Writing—review and editing. **Barrett M Powell**: Software; Investigation; Visualization; Methodology; Writing—review and editing. **Jia Jia Zhang**: Investigation; Methodology; Writing—review and editing. **Isabella Bolstad**: Investigation; Writing—review and editing. **Carolyn Vargas**: Investigation; Writing—review and editing. **Sandro Keller**: Resources; Supervision; Investigation; Project administration; Writing—review and editing. **Tania A Baker**: Supervision; Project administration; Writing—review and editing. **Robert T Sauer**: Conceptualization; Supervision; Funding acquisition; Investigation; Project administration; Writing—review and editing. **Joseph H Davis**: Conceptualization; Resources; Supervision; Funding acquisition; Investigation; Visualization; Methodology; Project administration; Writing—review and editing.

Source data underlying figure panels in this paper may have individual authorship assigned. Where available, figure panel/source data authorship is listed in the following database record: biostudies:S-SCDT-10_1038-S44318-025-00408-1.

## Disclosure and competing interests statement

The authors declare no competing interests.

