## [Peer Review File · The EMBO Journal]

An asymmetric nautilus-like HflK/C assembly controls FtsH proteolysis of membrane proteins

Alireza Ghanbarpour, Bertina Telusma, Barrett Powell, Jia Jia Zhang, Isabella Bolstad, Carolyn Vargas, Sandro Keller, Tania Baker, Robert T. Sauer, and Joseph Davis

Corresponding author(s): Joseph Davis (jhdavis@mit.edu), Robert T. Sauer (bobsauer@mit.edu), Alireza Ghanbarpour (alirezag@wustl.edu)

Review Timeline:

Submission Date:	28th Aug 24
Editorial Decision:	8th Oct 24
Revision Received:	28th Jan 25
Accepted:	28th Feb 25

Editor: Hartmut Vodermaier

Transaction Report:

Prof. Joseph Davis
Massachusetts Institute of Technology
77 Massachusetts Ave.
Cambridge, MA 02139

8th Oct 2024

Re: EMBOJ-2024-118876
An asymmetric nautilus-like HflK/C assembly controls FtsH proteolysis of membrane proteins

Dear Dr. Davis,

Thank you for submitting your study on the structure of native FtsH protease complexes to The EMBO Journal. It has now been seen by four referees with expertise in bacterial proteases, and in structural study of membrane proteins as well as of protease complexes. I am pleased to say that they all appreciate the quality of the work and the potential importance of its findings. Pending incorporation of various specific suggestions noted in the below-copied reports, we would therefore be happy to consider this work further for expedited publication.

Detailed information on preparing, formatting and uploading a revised manuscript can be found below and in our Guide to Authors; adhering to these guidelines as closely as possible shall greatly facilitate the editorial checking and processing at the time of resubmission - in particular regarding the completion of our author checklist, uploading of editable text files and individual figures, conversion of "supplemental" material into Expanded View and/or Appendix content, and inclusion of structured methods including a specific a Reagents and Tools Table (template download information below). Please also make sure to publicly deposit all generated dataset and make them available (at least for review) at the time of resubmission. Finally, I should remind you that it is our policy to allow only a single round of revision, making it important to carefully respond to all points raised at this point. As per our 'scooping protection' policy, related or competing work appearing during the course of the revision period will not affect our final decision on your study. Should you have additional questions linked to this decision, the referee reports, or the revision guidelines, please do not hesitate to contact me.

Thank you again for the opportunity to consider this study for The EMBO Journal! I look forward to receiving your revision in due time.

Yours sincerely,

Hartmut Vodermaier

9) To facilitate reproducibility and cross-laboratory adoption of methodologies, please structure the Materials & Methods section as outlined in our guide to authors, including a completed Reagents and Tools Table that can be downloaded from our author guidelines as well (<https://www.embopress.org/page/journal/14602075/authorguide#structuredmethods>).

10) Digital image enhancement is acceptable practice, as long as it accurately represents the original data and conforms to community standards. If a figure has been subjected to significant electronic manipulation, this must be clearly noted in the figure legend and/or the 'Materials and Methods' section. The editors reserve the right to request original versions of figures and the original images that were used to assemble the figure. Finally, we generally encourage uploading of numerical as well as gel/blot image source data; for details see: embopress.org/page/journal/14602075/authorguide#sourcedata

At EMBO Press, we ask authors to provide source data for the main manuscript figures. Our source data coordinator will contact you to discuss which figure panels we would need source data for and will also provide you with helpful tips on how to upload and organize the files.

In the interest of ensuring the conceptual advance provided by the work, we recommend submitting a revision within 3 months (6th Jan 2025). Please discuss the revision progress ahead of this time with the editor if you require more time to complete the revisions. Use the link below to submit your revision:

Link Not Available

Referee #1:

The FtsH-HflK/C complex was previously shown to form a symmetric bell-like closed structure with four copies of FtsH hexamer caged inside an outer shell consisting of 12 copies of HflK/C (Cell Research 32: 176-189 and Cell Reports 39). In this manuscript, Ghanbarpour et al present a new perspective on the structure-function of FtsH-HflK/C. They visualized the endogenous complex and found that FtsH-HflK/C is a partially open, asymmetric, nautilus-like structure with only 1 - 2 copies of FtsH hexamer. The authors went on to perform a series of experiments to make a compelling argument that their nautilus-like structure is likely the native one, and the previously published symmetric and closed structure could be an overexpression artefact. While I suspect the issue of the functional state of the complex - a closed cage vs nautilus - may not have been fully settled, the current work is a new perspective or thinking of the molecular mechanism of the system, and the work is systematic and very solid. The manuscript is clearly written, references properly cited, and the figures are well illustrated. I only have a few minor suggestions that do not require re-review:

1. The nautilus-like structure with fewer than 4 FtsH hexamers were also observed in the previous two Cell Reports papers but were considered as partial or ruptured states. This fact should be noted in the main text.
2. Suppl Fig. 5, bottom panels: the top view and side view labels should be exchanged.

3. I cannot find Fig. 4D: "FtsH catalyze lipid scrambling (Figure 4C-D)".
4. Defocus range in Table 1: "-0.05" most likely should be "-0.5"

Referee #2:

Ghanbarpour et al. present a beautiful nautilus-like structure of the HflKC-FtsH complex from *E. coli*, which may explain how HflKC modulates the activity of the FtsH protease. Interestingly, the newly presented cryo-EM structure differs notably in both shape and subunit composition from a previously published structure by Qiao et al. (2022). However, the authors offer several convincing explanations for these discrepancies and propose a revised model for the functionality of the complex. Furthermore, they conduct a global proteome analysis of the hflKC mutant, which supports a role of HflKC in the degradation of membrane proteins.

The manuscript is a pleasure to read. It is clearly written and beautifully illustrated. I have only a few minor comments.

1. Abstract, line 2: add "bacteria" to "assemblies in eukaryotic organelles". Otherwise, the abstract leaves the wrong impression that the HflKC-FtsH complex derives from organelles.
2. The authors suggest several possible explanations for the differences between their structure and the previously published one. A quick look into the Qiao paper revealed additional factors that could also contribute to the observed discrepancies between the two structures and might be worth discussing, such as point mutations in the Walker B and HExxH motifs of FtsH used to stabilize the cytosolic domain, as well as the application of chemical cross-linking.
3. Please follow standard nomenclature for genes and proteins, for example on page 6 the proteins should read SecY, DadA, YlaC, etc.

Referee #3:

This is an exciting manuscript which focuses on understanding the composition and function of the FtsH protease with HflK/C subunits in *E. coli*. Here they use a genomically tagged FtsH to purify a native complex and go on to solve the cryoEM of this complex revealing a novel nautilus like structure, with FtsH protease in the center and on opening gauged by the HflK/c subunits. This is in contrast to a previous cryoEM structure where the HflK/C subunits were closed. They see either one or two FtsH protease hexamer in the nautilus. They note negative curvature near the lipid bilayer portion, but notably do not observe density for lipid molecules. Using a proteoliposome assay they demonstrate lipid scramblase activity with stringent controls in place. Lastly, they perform a nice mass spectrometry analysis of the effect of HflK/C on protein abundance, using an FtsH pull down. They see changes in substrate association with FtsH in the absence of HflK, and surmise that the HflK/C subunits help to limit access of the substrate clientele, preventing degradation of large complexes.

Overall, this is a well written paper that adds to the field of proteostasis and membrane complex function. I have only a few comments for consideration:

1. Page 4. The curvature of the bilayer was difficult to see in the images presented. Maybe make this point more clear in figure and provide any references to support the outward (positive?) curvature of bacterial membranes. Are you sure this density is all lipid? Could detergent also be in this region? In the discussion, please address why lipids were not resolved.
2. In the proteomics data SecY and YlaC are both inner membrane proteins, however DadA is not TM -can the authors speculate on whether the nautilus potentially participate in proteostasis of periplasmic proteins?
3. For Extended Data Table 1, please add a legend to help readers recall the experiment conducted. Perhaps highlight those protein of interest in the table.
4. Extended Data Table 1 ylaC is not in the table.

Referee #4:

The paper presents the ex-vivo cryo-EM structure of the HflK/C - FtsH complex involved in the controlled degradation of badly folded non-functional membrane proteins. There are two recently published studies of this complex but the results and their interpretation somewhat differ in each of these studies. The structures presented here by Ghanbarpour et al, probably reflect a situation closer-to-native, with the complex being isolated without over expression and data being processed without applying any symmetry. The authors propose a role of HflK/C on changing membrane curvature and facilitating FtsH degradation activity on certain substrates, hinting at HflK/C's role as a modulator of FisH activity and supported by mass spec data. In general, the paper is very well written and illustrated, methods well reported and with a variety of assays and techniques

used. It is not clear to me the exact mechanism by which HflK/C would facilitate FtsH action on certain substrates rather than others. However, this can remain as a more-or-less open question to be addressed on further studies by this and other groups.

The authors properly acknowledge the work performed by other researchers, also commenting on the most recent discoveries. The discussions are honest and open. The work presented and the methodology is interesting for any structural biologist working on large and asymmetric complexes as well as for researchers working with membrane protein and membrane biophysics. There are not specific extra experiments that I can think of and that would strengthen substantially the current conclusions. Only some improvement could be obtained by minor additions. See below more itemised comments.

1. The maps were "just" autosharpened in Phenix. While this is just correct and sufficient, did the authors try any of the more advanced sharpening tools ?
2. Did the author try to perform additional symmetry expansion to gain details? This might be a lot of work for little gain of info)
3. In the tomography part, the authors could show an overlay of a tomogram slice with the reprojected volumes, to make Supp. Figure 6 more informative.
4. Even though the authors tried many different purification methods and reconstructions, often the resolution is limited and it is unclear to me whether the existence of C4 symmetric complexes as described by the other works can be really completely excluded. The complex could in principle exist in all different stoichiometric states.
5. Figure 1 could be clearer if the different domains of HflK/C proteins were indicated in the scheme. It is not easy for the reader to identify the SPFH domain in the figure.
6. How many times was the scramblase assay performed? This is not indicated in the figure caption not in the text.
7. Supplementary figure 1 could be clearer for the HflC and K parts
8. In the detergent-free purifications it is not so obvious that the nautilus has the same open shape.
9. While image processing is described in a very detailed way, model building is not. This part of the material and methods could be improved.

This paper is very well written and contains a lot of data. However, it is still to be elucidated the actual molecular mechanisms by which the HflK/C complex facilitates and regulates the degradation activity of FtsH. A natural question that comes to my mind is. Does FtsH bind more specifically to given nautilus curvatures?

Regardless, I think the paper can be published pretty much as it is, with some minor improvement.

We thank the reviewers for their comments. We have addressed all their points raised by in our revised manuscript, as detailed below.

REVIEWER COMMENTS

Referee #1:

The FtsH-HflK/C complex was previously shown to form a symmetric bell-like closed structure with four copies of FtsH hexamer caged inside an outer shell consisting of 12 copies of HflK/C (Cell Research 32: 176-189 and Cell Reports 39). In this manuscript, Ghanbarpour et al present a new perspective on the structure-function of FtsH-HflK/C. They visualized the endogenous complex and found that FtsH-HflK/C is a partially open, asymmetric, nautilus-like structure with only 1 - 2 copies of FtsH hexamer. The authors went on to perform a series of experiments to make a compelling argument that their nautilus-like structure is likely the native one, and the previously published symmetric and closed structure could be an overexpression artefact. While I suspect the issue of the functional state of the complex - a closed cage vs nautilus - may not have been fully settled, the current work is a new perspective or thinking of the molecular mechanism of the system, and the work is systematic and very solid. The manuscript is clearly written, references properly cited, and the figures are well illustrated. I only have a few minor suggestions that do not require re-review:

1. The nautilus-like structure with fewer than 4 FtsH hexamers were also observed in the previous two Cell Reports papers but were considered as partial or ruptured states. This fact should be noted in the main text.

We now note this fact on page 10 of the revised manuscript.

2. Suppl Fig. 5, bottom panels: the top view and side view labels should be exchanged.

We made this correction.

3. I cannot find Fig. 4D: "FtsH catalyze lipid scrambling (Figure 4C-D)".

This should have read (Figure 4B-C) and has been corrected.

4. Defocus range in Table 1: "-0.05" most likely should be "-0.5"

We have verified that the nominal defocus value of -0.05 μm is indeed correct. This microscope has an inherent defocus even nominally minimal defocus, so we sometimes use a nominal value of -0.05, which typically results in an actual defocus of approximately -0.3 μm .

Referee #2:

Ghanbarpour et al. present a beautiful nautilus-like structure of the HflKC-FtsH complex from *E. coli*, which may explain how HflKC modulates the activity of the FtsH protease. Interestingly, the newly presented cryo-EM structure differs notably in both shape and subunit composition from a previously published structure by Qiao et al. (2022). However, the authors offer several convincing explanations for these discrepancies and propose a revised model for the functionality of the complex. Furthermore, they conduct a global proteome analysis of the hflKC mutant, which supports a role of HflKC in the degradation of membrane proteins.

The manuscript is a pleasure to read. It is clearly written and beautifully illustrated. I have only a few minor comments.

1. Abstract, line 2: add "bacteria" to "assemblies in eukaryotic organelles". Otherwise, the abstract leaves the wrong impression that the HflKC-FtsH complex derives from organelles.

We made the suggested change.

2. The authors suggest several possible explanations for the differences between their structure and the previously published one. A quick look into the Qiao paper revealed additional factors that could also contribute to the observed discrepancies between the two structures and might be worth discussing, such as point mutations in the Walker B and HExxH motifs of FtsH used to stabilize the cytosolic domain, as well as the application of chemical cross-linking.

As suggested, we now note these possibilities on page 10 of the revised manuscript.

3. Please follow standard nomenclature for genes and proteins, for example on page 6 the proteins should read SecY, DadA, YlaC, etc.

We made these corrections.

Referee #3:

This is an exciting manuscript which focuses on understanding the composition and function of the FtsH protease with HflK/C subunits in E. coli. Here they use a genomically tagged FtsH to purify a native complex and go on to solve the cryoEM of this complex revealing a novel nautilus like structure, with FtsH protease in the center and on opening gauged by the HflK/c subunits. This is in contrast to a previous cryoEM structure where the HflK/C subunits were closed. They see either one or two FtsH protease hexamer in the nautilus. They note negative curvature near the lipid bilayer portion, but notably do not observe density for lipid molecules. Using a proteoliposome assay they demonstrate lipid scramblase activity with stringent controls in place. Lastly, they perform a nice mass spectrometry analysis of the effect of HflK/C on protein abundance, using an FtsH pull down. They see changes in substrate association with FtsH in the absence of HflK, and surmise that the HflK/C subunits help to limit access of the substrate clientele, preventing degradation of large complexes.

Overall, this is a well written paper that adds to the field of proteostasis and membrane complex function. I have only a few comments for consideration:

1. Page 4. The curvature of the bilayer was difficult to see in the images presented. Maybe make this point more clear in figure and provide any references to support the outward (positive?) curvature of bacterial membranes. Are you sure this density is all lipid? Could detergent also be in this region? In the discussion, please address why lipids were not resolved.

We added a citation (K. Khanna et al., Current Opinion in Structural Biology, 2022) that visualizes the outward curvature E. coli inner membrane using cryo-electron tomography on page 7 of the revised manuscript.

In the DDM-solubilized sample, the observed density could represent a mixture of lipids and detergent. However, in our DIBMA-reconstituted structure, no detergent was used; therefore, we expect all observed density to correspond to native lipids or the polymer itself, as the DIBMA polymer enables the extraction of membrane proteins along with native lipids.

We have amended the results (page 8) to note the possibility of detergent and lipid mixtures in our map.

2. In the proteomics data SecY and YlaC are both inner membrane proteins, however DadA is not TM -can the authors speculate on whether the nautilus potentially participate in proteostasis of periplasmic proteins?

DadA is reported as a inner membrane associated protein (<https://pubmed.ncbi.nlm.nih.gov/7906689/>; <https://pubmed.ncbi.nlm.nih.gov/6118175/>; <https://pubmed.ncbi.nlm.nih.gov/6102989/>; <https://pubmed.ncbi.nlm.nih.gov/6133564/>), which could tether it in proximity of the HflKC/FtsH complex and, because HflK has a cytoplasmic

extension, it is possible that it affects degradation of inner-membrane-associated cytoplasmic proteins.

3. For Extended Data Table 1, please add a legend to help readers recall the experiment conducted. Perhaps highlight those protein of interest in the table.

The following caption was added, with proteins noted in the text highlighted: **Extended Data Table 1:** Proteins pulled down by FtsH from wild-type or Δ hflK/C cells, with eluates analyzed by mass spectrometry.

4. Extended Data Table 1 ylaC is not in the table.

We were unable to identify YlaC in our pull-down experiment, which may be due to factors such as how strongly it assembles with FtsH or the FtsH-HflK/C complex. However, our proteomic analysis revealed changes in its levels, and previous studies have reported YlaC as a substrate of FtsH (J. Arends et al., Proteomics, 2016).

Referee #4:

The paper presents the ex-vivo cryo-EM structure of the HflK/C - FtsH complex involved in the controlled degradation of badly folded non-functional membrane proteins. There are two recently published studies of this complex but the results and their interpretation somewhat differ in each of these studies. The structures presented here by Ghanbarpour et al, probably reflect a situation closer-to-native, with the complex being isolated without over expression and data being processed without applying any symmetry. The authors propose a role of HflK/C on changing membrane curvature and facilitating FtsH degradation activity on certain substrates, hinting at HflK/C's role as a modulator of FisH activity and supported by mass spec data.

In general, the paper is very well written and illustrated, methods well reported and with a variety of assays and techniques used. It is not clear to me the exact mechanism by which HflK/C would facilitate FtsH action on certain substrates rather than others. However, this can remain as a more-or-less open question to be addressed on further studies by this and other groups.

The authors properly acknowledge the work performed by other researchers, also commenting on the most recent discoveries. The discussions are honest and open. The work presented and the methodology is interesting for any structural biologist working on large and asymmetric complexes as well as for researchers working with membrane protein and membrane biophysics. There are not specific extra experiments that I can think of and that would strengthen substantially the current conclusions. Only some improvement could be obtained by minor additions. See below more itemized comments.

1. The maps were "just" autosharpened in Phenix. While this is just correct and sufficient, did the authors try any of the more advanced sharpening tools ?

We did not use advanced sharpening tools such as LocScale, primarily because the resolution of most maps was not high enough to justify the use of these tools that, in our hands, tend to over-sharpen such medium-resolution maps. While not a "advanced", we note in the methods section that a subset of maps were sharpened using the map sharpening tool from cryoSPARC.

2. Did the author try to perform additional symmetry expansion to gain details? This might be a lot of work for little gain of info).

Symmetry expansion is typically useful when particles show some form of symmetry, either true or pseudo. The nautilus-like assembly is not pseudo-symmetric; therefore, we do not believe performing symmetry expansion would be helpful.

3. In the tomography part, the authors could show an overlay of a tomogram slice with the reprojected volumes, to make Supp. Figure 6 more informative.

We have added this representation as suggested.

4. Even though the authors tried many different purification methods and reconstructions, often the resolution is limited and it is unclear to me whether the existence of C4 symmetric complexes as described by the other works can be really completely excluded. The complex could in principle exist in all different stoichiometric states.

We appreciate reviewer's comments. However, we do not believe resolution limits visualization of nautilus structure (presented here) vs. cage structure. As shown below using low pass filtered maps of two structures even in resolution as low as 20 Å the differences between the two types of assemblies are evident. Moreover, deep classification of these structures using cryoDRGN or 3D classification as implemented in cryoSPARC failed to reveal structures consistent with the closed cage-like structure.

5. Figure1 could be clearer if the different domains of HflK/C proteins were indicated in the scheme. It is not easy for the reader to identify the SPFH domain in the figure.

We appreciate the comment. We have included the AlphaFold model of HflK and HflC, incorporating some unstructured regions that were not modeled in our own structures or in previous cage structures with defined domain organization in Supplementary Figure 1.

6. How many times was the scramblase assay performed? This is not indicated in the figure caption not in the text.

The assay was done three times. We have added this information in the figure caption.

7. Supplementary figure 1 could be clearer for the HflC and K parts.

We have addressed this, as noted above (see response to point 5).

8. In the detergent-free purifications it is not so obvious that the nautilus has the same open shape.

Although the resolution of this DIBMA-reconstituted complex is lower than that of other structures, it reveals the overall features of detergent-solubilized maps, with an open nautilus-like assembly and two FtsH hexamers per assembly as shown in Figure 3. As the reviewer correctly points out, we see some flexibility in the opening of the nautilus both within (as analyzed by cryoDRGN) and across datasets/reconstitution methods. We believe these changes reflect inherent flexibility in and around the nautilus opening.

9. While image processing is described in a very detailed way, model building is not. This part of the material and methods could be improved.

We have provided additional explanation for the model building in the methods section.

This paper is very well written and contains a lot of data. However, it is still to be elucidated the actual molecular mechanisms by which the HflK/C complex facilitates and regulates the degradation activity of FtsH. A natural question that comes to my mind is. Does FtsH bind more specifically to given nautilus curvatures?

We appreciate the reviewer's comment. We agree that there are additional questions to be addressed, but our structures may provide a step forward in generating new hypotheses suggested by these structures, which can be tested in future work.

Regardless, I think the paper can be published pretty much as it is, with some minor improvement.

Prof. Joseph Davis
Massachusetts Institute of Technology
Biology
77 Massachusetts Ave.
Cambridge, MA 02139

28th Feb 2025

Re: EMBOJ-2024-118876R
An asymmetric nautilus-like HflK/C assembly controls FtsH proteolysis of membrane proteins

Dear Dr. Davis,

Thank you for submitting your final revised manuscript for our consideration. I am pleased to inform you that in light of the positive re-review by one of the original referees (below), we have now accepted it for publication in The EMBO Journal.

Yours sincerely,

Hartmut Vodermaier

Referee #4:

This was the second round of review. After reading the revised paper and the answers to reviewers, I am satisfied with the current version of the manuscript and I would not require any major change. A very minor thing: I would adjust the supplementary video and cut it shorter.